# Documentary Analysis of *Hypericum perforatum* (St. John’s Wort) and Its Effect on Depressive Disorders

**DOI:** 10.3390/ph17121625

**Published:** 2024-12-03

**Authors:** María Carolina Otero, Francisco Ceric, Sebastián Miranda-Rojas, Carolina Carreño, Rachelly Escares, María José Escobar, Chiara Saracini, Cristian Atala, Ricardo Ramírez-Barrantes, Felipe Gordillo-Fuenzalida

**Affiliations:** 1School of Chemistry and Pharmacy, Faculty of Medicine, Andrés Bello University, Santiago 8370146, Chile; 2Laboratory of Affective Neuroscience, Faculty of Psychology, Universidad del Desarrollo, Santiago 7610658, Chile; fceric@udd.cl; 3Department of Chemical Sciences, Faculty of Exact Sciences, Andrés Bello University, Santiago 8370146, Chile; sebastian.miranda@unab.cl; 4Center for Theoretical & Computational Chemistry (CQT&C), Department of Chemical Sciences, Faculty of Exact Sciences, Andrés Bello University, Santiago 8370146, Chile; 5School of Medical Technology, Faculty of Medicine, Andrés Bello University, Santiago 8370146, Chile; 6The Neuropsychology and Cognitive Neurosciences Research Center (CINPSI Neurocog), Faculty of Health Sciences, Universidad Católica del Maule, Avenida San Miguel 3605, Talca 3460000, Chile; 7Instituto de Biología, Facultad de Ciencias, Pontificia Universidad Católica de Valparaíso, Valparaíso 8331150, Chile; 8School of Medical Technology, Faculty of Medicine, Andrés Bello University, Viña del Mar 8370035, Chile; ricardo.ramirez@unab.cl; 9Laboratory of Applied Microbiology, Center for Biotechnology of Natural Resources, Faculty of Agrarian and Forestry Sciences, Catholic University of Maule, Avda. San Miguel 3605, Talca 3460000, Chile; fgordillo@ucm.cl

**Keywords:** medicinal plants, naphthodianthrones, phloroglucinols, antidepressants

## Abstract

*Hypericum perforatum*, also known as St. John’s Wort, pericon, or yellow grass, is known for its antidepressant potential. It could represent a natural alternative to current pharmacological antidepressant treatments, which have a high incidence of side effects in patients and therefore lead to early dropouts. Through a bibliographic revision of clinical trials and information collected from scientific articles during the first period of 2020, we aimed to evaluate whether its administration could be beneficial in the treatment of mild-to-moderate depression, with fewer side effects compared to synthetic drugs. Among the main components, hypericin and hyperforin have been related to the observed antidepressant activity; therefore, their possible mechanism of action was reviewed and highlighted. Furthermore, patients receiving *Hypericum* extracts were less likely to withdraw from studies because of adverse effects compared to those receiving older standard antidepressants. This review aims to provide suggestions for an alternative treatment of mild-to-moderate depression disorder under the supervision of a medical doctor, since, although it appears to be a potentially efficient treatment with a low presence of adverse effects in comparison to synthetic antidepressants, it might also interact with other medications and lead to therapeutic failures if misused for self-medication.

## 1. Introduction

Depression is one of the most common disorders in the world, affecting approximately 280 million people worldwide [1]. The World Health Organization (WHO) defines it as “a frequent mental disorder, characterized by the presence of sadness, loss of interest or pleasure, feelings of guilt or lack of self-esteem, sleep or sleep disorders, lack of appetite, feeling tired, and lack of concentration” [2]. In addition to the symptoms described in the WHO definition, we can also find psychological symptoms. In the Diagnostic and Statistical Manual of Mental Disorders, fifth edition (DSM-V) [3], depression is defined as “Feelings of sadness, low mood, and loss of interest in their usual activities must mark a change from a person’s previous level of functioning and have persisted for at least two weeks”, and can be diagnosed if at least five of the following symptoms are persistently present: change in appetite, and the losing or gaining of weight; sleeping too much or not sleeping well (insomnia); fatigue and low energy on most days; feeling worthless, guilty, and hopeless; the inability to focus and concentrate, which may interfere with daily tasks at home, work, or school; movements that are unusually slow or agitated (a change which is often noticeable to others); thinking about death and dying; suicidal ideation or suicide attempts. These can be associated with altered states such as anxiety, emotional emptiness, negative thoughts, memory loss, delusions, and hallucinations, along with physical symptoms such as weight gain or loss [4].

An estimated 3.8% of the population experience depression, including 5% of adults (4% among men and 6% among women) and 5.7% of adults older than 60 years [5]. Three main levels are identified by its categorization, mild, moderate, and severe, where the classification within any of these levels will depend on the intensity of the symptoms, its duration in time, and the functional deterioration that the patient presents [6].

If depression was already acknowledged as one of the most diffuse clinical conditions, after the SARS-CoV-2 pandemic in 2020, there has been a worrying increase in depressive symptoms in both men and women, particularly in the symptoms of anxiety and/or depression, which are associated with negative feelings, social isolation, and, in the case of university students, an interruption of their studies [7,8].

Treatment options encompass pharmacotherapy, particularly in the acute phase, with the goal of achieving remission or, ideally, restoring the individual to their baseline level of functioning. Additional approaches include psychotherapy, cognitive and behavioral therapies, and, more recently transcranial magnetic stimulation (TMS), and transcranial direct current stimulation (tDCS). Furthermore, deep brain stimulation (DBS) and vagus nerve stimulation (VNS) are also utilized, each exhibiting varying degrees of efficacy [9].

The antidepressant drugs used today can be classified into six groups depending on their mechanism of action, either by inhibiting the enzymes that degrade catecholamines or by interacting with their transporters to increase their availability. These groups are defined as follows: selective serotonin reuptake inhibitors (SSRIs), tetracyclic/unicyclic drugs, serotonin–noradrenaline reuptake inhibitors (SNRIs), serotonin receptor modulators, tricyclic antidepressants, and monoamine oxidase inhibitors (MAOIs) [10]. The problem with these synthetic antidepressants is that the most common side effects negatively impact the health of patients, including nausea, increased appetite and weight gain, fatigue, constant sleep, insomnia, dry mouth, constipation, dizziness, vertigo, anxiety, agitation, cardiac problems, and genetic variations.

Considering the number of adverse effects that conventional synthetic antidepressants have, despite their evolution, the search for a natural alternative with fewer side effects to replace these drugs is still a valid option. St. John’s Wort (*Hypericum perforatum*) manages to stand out among medicinal plants, since it is known to have the potential to act as an antidepressant and sleep-inducer [11].

*H. perforatum* is part of the ethnopharmacology used in European regions to treat depression and anxiety, being one of the most prescribed treatments for depression in Germany [12]. This plant has been shown to be more effective than placebos [13] in the treatment of depression, and its side effects are estimated to be much less than those of conventional antidepressant therapies [14]. Many reviews since the early 2000s have highlighted the efficiency of *H. perforatum* in diminishing depressive symptoms [15,16].

Its mechanism of action, although proposed as assimilating to those tricyclic antidepressants and selective serotonin reuptake inhibitors [17], has not yet been well established (Table 1).

In this review, we explore and investigate the components of *H. perforatum*, its potential effects on depression treatment, and the proposed mechanism of action, in addition to its possible adverse effects. This was achieved by obtaining different *H. perforatum* extracts, of which those labeled as ZE 117, STW3, LI 160, W5579, and W5579 stood out according to their hypericin and hyperforin contents (Table 2).

This review highlights the efficiency of *H. perforatum* extracts and the low presence of adverse effects that they produce in patients, highlighting these attributes in the extract labeled as ZE 117, which contains between 1 and 4% of hyperforin from the total extract of the herb, as well as a negligible amount of hypericin. Although the extracts of this herb are effective for mild-to-moderate depressive disorders, co-administration with other antidepressant drugs could lead to the therapeutic failure of the drugs, and, thus, the detriment of the treatment of the psychiatric disease.

## 2. Materials and Methods

A review of studies was conducted by gathering information from articles and clinical trials sourced from scientific publication platforms to determine the efficacy of St. John’s Wort in treating depression and to explore its potential mechanisms of action.

For the search and collection of information, different platforms were used, such as “Web of Science”, “Scopus”, “PubMed”, and “SciELO”, among others, related to scientific dissemination, topics such as the constituents associated with the herb are discussed, its mechanisms of action, and its relationship with mild-to-moderate depression, as well as a comparison of the adverse effects of *H. perforatum* with synthetic antidepressants, and the interactions of *H. perforatum* with drugs intended for other pathologies, were investigated. The collection of sixty publications comprised scientific papers, experimental studies, and other works that fulfilled the established inclusion and exclusion criteria, allowing for the effective segregation of the gathered information.

Inclusion criteria: Experimental studies published within the last 24 years (from 2000 to 2024) were included, while the most recent clinical guidelines or protocols for the treatment of depression were prioritized. If recent guidelines were unavailable, those no older than 10 years were chosen instead.

Exclusion criteria: Experimental studies that were published more than 24 years ago; studies that included participants under 18 years of age, over 80 years of age, and/or pregnant women.

The following keyword strings were used to perform the searches in the main scientific databases (Web of Science, Scopus, PubMed, and SciElo). We found 945 results, which have been revised with the following inclusion criteria in mind: experimental studies must be a maximum 24 years old (at earliest 2000); the clinical protocols used in the studies should be the most recent (not exceeding 10 years, so at earliest 2010). From all of the discussed articles, only 16 articles included randomized controlled trials comparing the effectiveness of *H. perforatum* with other medications, including standard antidepressants, in the treatment of depression. Amongst the ones which passed the first phase, we excluded studies on under-18 and over-80 participants, and/or pregnant women.

## 3. *Hypericum perforatum* and Its Effect on Depressive Disorders

### 3.1. Main Characteristics of the Plant

*Hypericum perforatum* is a perennial herb with glabrous leaves and distinct yellow flowers with black glands along the margin of the petals. The species belongs to the genus *Hypericum*, the most diverse genus in the Hypericaceae, including over 500 species [22,23,24]. *H. perforatum* is native to temperate regions of the Northern Hemisphere, particularly Eurasia and parts of Africa [23]. This plant currently has a worldwide distribution (except the poles), growing in continental Chile from Valparaíso to the Los Lagos region, being also present in the Juan Fernandez archipelago [24]. The first known record in Chile dates from 1869, and it is considered to be highly invasive [25,26]. This hemicryptophyte plant produces capsules but can also reproduce vegetatively [27]. Its success and spread may partially be due to its ability to self-pollinate and to form apomictic seeds [28]. This plant can hybridize, and there are several cultivars and varieties that can significantly vary in their chemical composition [29,30].

### 3.2. Chemical Composition of H. perforatum

Extracts from *H. perforatum* presented several therapeutic effects, which have increased the interest and scientific relevance of this species and its bioactive molecules [31]. The therapeutic effects which have been reported for the plant are attributed to several compounds already identified, which have contributed to its antidepressive and anxiolytic properties [32,33,34]. However, other properties of the extracts, such as antimicrobial, antinociceptive, antioxidant, antidiabetic, wound-healing, and anti-inflammatory properties, have also been reported [35,36,37]. Several compounds have been identified in the plant, such as phenolic compounds (naphthodianthrones, phloroglucinols, flavonoids, bioflavonoids, proanthocyanidins, and tannins), hydrocarbons, phytoesterols, and volatile oils, among others. In this section, we compiled information on some of the components found in high concentrations within the plant compared to other molecules, such as naphthodianthrones and derivatives of phloroglucinol, which are suggested to be the bioactive compounds responsible for the observed antidepressive and anxiolytic activities [33] (Figure 1).

Naphthodianthrones are a group of phenolic compounds (diantronic derivatives) composed of molecules such as protohypericin, hypericin, protopseudohypericin, and pseudohypericin. The levels of pseudohypericin and its precursor, hypericin, were analyzed in *H. perforatum* and various species of Hypericum plants found on the island of Crete, Greece. Interestingly, hypericin was found in all of the species, where the highest concentration was found during blossoming and the lowest content was present during the ripening of the fruit. Additionally, *H. perforatum* showed high hypericin levels in the flowers and fruits compared to the other parts of the plant, such as the leaves and petioles. In general, the total amount of hypericins in *H. perforatum* (protohypericin, hypericin, protopseudohypericin, and pseudohypericin) per the entire aerial part of the plant was calculated for all developmental stages, where it was reported that the plant contained the highest amount of hypericin compared to other species. Moreover, this accumulation was dependent on the altitude [38,39]. In other work, naphthodianthrones, phloroglucinols, flavonoids, and biflavones were identified in the hydroalcoholic extracts (60% ethanol or 80% methanol) of the aerial part of the plant. Hypericin and pseudohypericin were found in concentrations from 0.03% to 0.30% of the dry weight, showing limited solubility in almost all of the solvents and varying according to the stage of development of the herb. Hyperforin was found to be present between 2.00% and 4.50% of the main phloroglucin content; meanwhile, other flavonoid compounds comprised an average of 2.00 to 4.00% [37].

To establish the composition of the ethanolic extracts of the flowers from *H. perforatum* obtained from different altitudes, high-performance thin-layer chromatography (HPTLC) was used to identify the metabolites hypericin, quercetin, and chlorogenic acid. In this work, the total phenolic content was also reported, showing a range between 27.87 and 41.77 mg/g [40]. Furthermore, in aqueous fractions extracted from the aerial parts and fractionated with different solvents, 0.03–0.13 mg/g of hypericin, 0.39–54.68 mg/g of hyperforin, 0.01–0.06 mg/g of catequin, and 0.28–2.57 mg/g of chlorogenic acid, among others, were obtained [41]. Other study analyzed extracts from the flowers, leaves, and roots, obtaining different values depending on the compound. Values of 1.04–3.69 mg/g of hypericin were obtained in the flowers, 0.27–0.62 mg/g in the leaves, and 21.6 mg/g in the roots; meanwhile, values of 28.13–41 mg/g of hyperforin were obtained in the flowers, 15.44 mg/g in the leaves, and 0.32 mg/g in the roots, while flavonoids such as catechin were obtained at values of 4.64 in the flowers, 2.46 in the leaves, and 4.57 mg/g in the roots [42].

### 3.3. Mechanism of Action

As mentioned before, although the exact mechanism behind the antidepressant activity of *H. perforatum* is not fully understood, it appears to involve multiple pathways [43]. These pathways are primarily attributed to its bioactive compounds, such as hyperforin, hypericin, and various flavonoids. In general terms, these mechanisms include the inhibition of neurotransmitter reuptake, which affects neurotransmitters like serotonin, noradrenaline, dopamine, GABA, and L-glutamate [44,45]. This inhibition leads to increased levels of these neurotransmitters in the synaptic cleft, potentially contributing to its antidepressant effects by acting similarly to selective serotonin reuptake inhibitors (SSRIs) [46].

The molecular mechanisms through which hyperforin exerts its effects are not clearly understood. However, one hypothesis suggests that this is primarily due to the activation of TRPC6 channels [47,48,49]. TRPC6 channels are expressed in the hippocampus as well as in other cortical regions associated with depression [50,51,52,53]. Additionally, it is well documented that TRPC6 plays a role in synaptic plasticity responses in neurons, including dendritic growth, axonal sprouting, spine morphology, and an increase in excitatory synapses [51,54,55,56]. Interestingly, the deficit of TRPC6 in mice produces anxious and depressive behaviors while also reducing the excitability of hippocampal CA1 pyramidal neurons [47].

The role of hyperforin on these channels may be similar to that of the brain-derived neurotrophic factor (BDNF), as it modulates the activity and expression of the channel [51,57]. Additionally, it has been shown that hyperforin possesses a specific binding motif in the TRPC6 C-terminus, which is crucial for hyperforin sensitivity; mutations in this motif abolish hyperforin’s ability to activate TRPC6. Furthermore, hyperforin does not activate other closely related TRPC channels such as TRPC3 and TRPC7, strongly suggesting that hyperforin is a selective activator of TRPC6 [47].

The relationship between neuronal excitation and antidepressant action is complex and likely involves a cascade of downstream effects. Increased neuronal activity may influence the release of neurotransmitters like serotonin and dopamine, which are known to play a crucial role in mood regulation. Additionally, this activity could impact neuroplasticity, which is thought to be impaired in depression [43,46,58,59].

Moreover, it may affect neurotransmitter reuptake. Sell et al. demonstrated in several in vitro neuronal models that the activation of TRPC6 functions as a protonophore, causing cytosolic acidification, which fuels the plasma membrane sodium–proton exchanger [49]. These effects lead to an increase in the free intracellular sodium concentration and inhibit neurotransmitter uptake via Na^+^ cotransport. Moreover, this activation depletes large dense core vesicles, which require a pH gradient to accumulate monoamines, potentially influencing neurotransmitter release [49]. To gain a more comprehensive understanding, further research into the downstream effects of TRPC6 activation, and how these effects relate to the mechanisms of antidepressant action, would be beneficial.

In this context, *H. perforatum* has been suggested to function similarly to selective serotonin reuptake inhibitors (SSRIs) because it appears to inhibit the synaptosomal reuptake of serotonin (and dopamine) in the brain [60].

Specifically, regarding depression, it has been reported that 5-HT1A autoreceptors may be overactive, leading to reduced serotonin release and contributing to depressive symptoms [61,62]. Previous studies suggest that St. John’s Wort can restore the decrease in 5-HT1A receptor binding induced by chronic mild stress and significantly increase 5-HT1A receptor binding in the frontal cortex and hippocampus of rats, leading to an antidepressant effect [63]. Moreover, the inhibition of serotonin reuptake has been reported to occur due to an increase in intracellular free sodium ions through the activation of nonselective cation channels such as TRPC6 [64,65]. This mechanism allows more serotonin to remain available in the brain, which can have mood-boosting effects.

Despite this evidence, it is important to note that St. John’s Wort is a complex herbal supplement, and its effects on serotonin may not be as straightforward or predictable as those of selective serotonin reuptake inhibitors (SSRIs) [43].

On the other hand, there is evidence that compounds in *H. perforatum* may also modulate monoamine oxidase (MAO) inhibition. Some studies suggest that various components, including hypericin, hyperforin, quercetin, and flavonoids, may inhibit both the MAO-A and MAO-B, enzymes, which are responsible for breaking down monoamine neurotransmitters such as serotonin, norepinephrine, and dopamine. However, the effects on dopamine and norepinephrine are less conclusive, and the molecular mechanisms involved remain largely unknown [44,61,66,67].

Additionally, more general mechanisms have been associated with the modulation of the hypothalamic–pituitary–adrenal (HPA) axis. Hyperactivity of the HPA axis is sometimes indicative of depression [68] and is modulated by an increase in the corticotropin-releasing factor (CRF) [69]. Notably, the selective antagonism of CRF genes has been reported in hypericin and pseudohypericin [69,70,71].

Based on the available research, *H. perforatum* appears to have some efficacy in treating mild-to-moderate depression. Studies suggest that it can improve mood and reduce depressive symptoms, potentially by influencing neurotransmitter systems, particularly serotonin and dopamine [43]. It has been highlighted that St. John’s Wort might be as effective as some SSRIs in treating mild-to-moderate depression.

However, it is crucial to emphasize that the mechanisms are not fully understood. As with any treatment, individual responses to *H. perforatum* can vary, and it may not be effective for everyone. Additionally, St. John’s Wort can interact with various medications, including antidepressants, birth control pills, and blood thinners. Therefore, it is essential to consult with a healthcare professional before using *H. perforatum*, especially in combination with other medications or in those with underlying health conditions.

### 3.4. Comparison of the Adverse Effects of the Consumption of St. John’s Wort with Those of the Consumption of Synthetic Antidepressants

After a revision of several studies, and under the following inclusion parameters: patients with average Hamilton scale HAMD-17 (15–20 points) corresponding to mild–moderate depression, an average age of 42 years (with all being older than 18 years), and who have undergone treatment with an average duration of 6 weeks, a global population of 3599 participating patients was obtained from these studies, of which 2004 of the patients were treated with extracts of *H. perforatum* (Hp) (56%), 877 were treated with synthetic antidepressants (24%), and 718 were treated with a placebo (20%) on a daily basis. To measure the antidepressant efficacy in these three groups, the term drug response is used to refer to the percentage of patients presenting a reduction of at least 50% on the HAMD-17 scale so to compare their efficacy.

Patients treated with *H. perforatum* mostly consumed ZE 117, W5579, or STW3 extracts in doses ranging from 300 to 1200 mg orally daily in capsules. These extracts differed in their concentrations of hypericin and hyperforin, as observed from the data listed in Table 3.

There are some studies that treated patients with synthetic antidepressants in parallel to *H. perforatum* so to obtain a comparison between both treatments. Among these studies, those treated with imipramine, fluoxetine, sertraline, and citalopram are noted. These results are summarized in Table 4, where it is observed that SSRIs are the drugs of choice for the treatment of mild-to-moderate depression, corresponding to 81% of the treatments administered to a total of 877 patients, who took these medications at their prescribed dosages and successfully completed their treatment protocols within the studies. The most-consumed drug in the studies was fluoxetine, followed by sertraline, imipramine, and, finally, citalopram, the latter being administered in the lowest dose, at only 20 mg daily, for an average of six weeks.

A synthesis of what was obtained in all of the studies is presented in Figure 2, where the data obtained from *H. perforatum* and its different extracts are compared with the results of the synthetic drugs and placebos using the concept of responders. The data are presented as the percentage of patients who, when consuming the treatment for the indicated time, showed a decrease in the HAMD-17 scale of at least 50% of their initial assessment. This allows for the identification of which treatments were better received and led to greater improvements among the patients who used them. This calculation was made according to the following formula:$$\mathrm{Frequency}\;(\%)=\frac{No.\;of\;responding\;patients\;for\;this\;treatment\;\times \;100}{Total\;number\;of\;people\;with\;this\;treatment}$$

In the results obtained, ZE 117 stands out within the *H. perforatum* extracts and imipramine stands out within the synthetic components, with a higher frequency of responders.

The participants treated with the *Hypericum* extracts demonstrated a lower likelihood of discontinuing the studies due to adverse effects compared to those treated with older standard antidepressants (OR = 0.24; 95% CI: 0.13–0.46; I^2^ = 0%) or SSRIs (OR = 0.53; 95% CI: 0.34–0.83; I^2^ = 0%) [87].

Despite the low percentage of side effects that the *H. perforatum* extracts presented, they were different when comparing the parameters between the extracts themselves, with an oscillation between 3% and 21% of differences between them.

In the patients treated with *H. perforatum*, the extracts that presented the highest side effects were STWE3 and W5570, with 38% and 34% of the patients, respectively. The ZE117 extract presented fewer of these effects, at only 19% among its study population. Although the intensity of the side effects presented by the patients is essential when comparing one treatment with another, so is the type of side effect presented, since mild-to-moderate depression has a wide range of them.

On the other hand, Figure 3 shows the percentage of the incidence of adverse effects associated to synthetic antidepressants, where the presence of side effects in patients was high. From this, imipramine showed the highest incidence, with 63%, followed by citalopram, with 42%, and, lastly, sertraline, with 32%, which, despite being the lowest, still has a significant percentage of adverse effects.

Some of the side effects evidenced from the revised studies with synthetic antidepressants included sexual dysfunction, agitation, joint pain, and anxiety; none of these were experienced by the users of *H. perforatum*.

Regarding the side effects that presented the highest frequency in both groups, namely, those treated with extracts of *H. perforatum* and those treated with synthetic antidepressants, we can find general malaise, gastrointestinal disorders, colds, drowsiness, diarrhea, respiratory diseases, and abdominal pain.

In Table 5, a comparison is presented between the study title and key findings on the side effects of the treatments with *H. perforatum* and synthetic drugs. Moreover, Table 6 shows a comparative table of the side effects of *H. perforatum* vs. synthetic/traditional antidepressants. From the table, we can conclude that *H. perforatum* generally exhibits a more favorable side-effect profile compared to synthetic and traditional antidepressants. The table indicates that patients treated with *H. perforatum* experience lower incidences of common side effects such as nausea, fatigue, dry mouth, dizziness, and sexual dysfunction. While certain side effects, such as insomnia and gastrointestinal issues, occur similarly with both treatments, many adverse effects are significantly less frequent with *H. perforatum*. This suggests that *H. perforatum* may be a safer alternative for individuals concerned about the side effects associated with conventional antidepressant therapies. However, it is important to consider the overall efficacy and individual patient circumstances when making treatment decisions.

## 4. Discussion

Mental health in Chile evidences important aspects that need to be improved. One of the most relevant aspects corresponds to the treatment of depression, since 70% of the Chilean population currently claims to have or to have had a mental problem or illness. Despite the great variety of existing drugs, they do not always satisfy the patient’s needs in the most efficient way, since they usually generate a significant number of adverse effects, potentially leading to dropout from the treatment, which is why others have sought alternatives that present fewer side effects. In this context, *H. perforatum* stands out as a natural option that is used for its potential antidepressant capacity. The foundational studies used to inform the treatment of depression in Chile are not sufficiently representative, as they are conducted in diverse populations and countries, predominantly in European nations that do not closely resemble the national context of Chile. Furthermore, these studies are rarely updated, and there is a notable absence of rigorous scientific research conducted in Chile regarding the effectiveness of herbal treatments for these mental disorders.

In this review, we have identified the impact of the daily dosing of *H. perforatum* extracts on mild-to-moderate depressive disorders, as well as significant herb–drug interactions. Together, these data highlight the importance of evaluating the nature of the components of *H. perforatum* that are related to antidepressant effects, such as herb–drug interactions as a critical component with respect to the likelihood of a significant interaction.

*H. perforatum* has been identified as an effective and profitable alternative to conventional antidepressants, since it is a plant that does not require many conditions when it comes to crop growing, in addition to its lower commercial value when compared to synthetic antidepressants [97,98,99].

Among its components, hypericin and hyperforin exhibit the highest potential for antidepressant activities due to their greater concentrations. Research has partially delineated their mechanisms of action, particularly in mediating the broad-spectrum inhibition of neurotransmitter reuptake, in particular serotonin, dopamine, noradrenaline, glutamate, and gamma-aminobutyric acid [100,101]. These components also mediate the apparent reduction in intestinal absorption and drug bioavailability by inducing the expression of the intestinal multidrug resistance gene 1 (MDR1 and ABCB1), intestinal and hepatic CYP3A4, among other cytochrome isoforms [102].

In addition, the plant has many other components, such as flavonoids (hyperoside, isoquercitin, rutin, and quercetin), biflavonoids (I3′II8-biapigenin and amentoflavone), proanthocyanidins (isoflavones), and chlorogenic acid (caffeic acid, ferulic acid and p-coumaric acid), which are found in a lower concentration than in hypericin and hyperforin. Even though some of these have been observed to have an antidepressant capacity, such as proanthocyanidins in cranberry [25] or chlorogenic acid in coffee [103], where they are found in a greater quantity, this has not been established for *H. perforatum* due to its low concentrations. Thereby, there is a lack of evidence to determine if, in the low concentrations in which they are found in *H. perforatum*, these compounds are useful for an antidepressant treatment. In addition, it would still be necessary to carry out in vitro and in vivo studies to explore if the components would have some type of interaction, either enhancing or inhibiting each other, and thus being able to establish the best approach to use of the plant and to enhance its effect.

Despite *H. perforatum* having an antidepressant effect with low side effects compared to the synthetic drugs used for the same purpose, the totality of its therapeutic scope and the level of interactions that its consumption can produce, especially when compared to other drugs, suggests the need for careful consideration and professional guidance before its use. This is particularly important in patients who are taking multiple medications or that have underlying health conditions, as potential interactions could outweigh the benefits of the herbal treatment [87].

Various studies have shown that the extracts WS 5571 (900 mg), STW3-IV (900 mg), WS 5570 (600 mg), PM 235 (500 mg), and ZE 117 (500 mg) of *H. perforatum*, with their daily dosages, respectively, demonstrated an antidepressant effect in patients with an average of six weeks of consumption, which was represented in an average of 66% of responders based on the Hamilton scale used in these studies [13,77,81,82,104]. Along with this, these investigations reflected the good tolerance to the extracts by the participants, with only 22% of them presenting some adverse effect, evidencing their effectiveness as effective and safe extracts for patients with depressive disorders.

Regarding the comparison of the antidepressant effectiveness of *H. perforatum* extracts and synthetic drugs, a study developed by [75] compared the effectiveness of omipramine for an average of 6 weeks with 500 mg daily of the ZE 117 extract for the same period. Based on the parameters of the responders associated with the Hamilton scale, they concluded that both had the same antidepressant effect, but that the herbal extract proved to be safer by presenting fewer side effects in its study population (63% vs. 39%) [75]. Following the previous methodology, in the same year, Schrader et al. (2000) compared fluoxetine (20 mg daily) with a dose of 500 mg daily of the ZE 117 extract [74], which was carried out during one year, as observed in Table 5 [73], and in later years by other researchers, who maintained the comparison of the drug and the dosage; an equivalent study was carried out with a daily dose of 900 mg of the LI 160 extract. Based on these studies, a similar antidepressant efficacy was concluded for both of the treatments, with a better tolerance in the case of the *H. perforatum* extract, as can be seen in Figure 2. The same procedure was repeated during later years in the comparison of both sertraline (50 mg) and citalopram (20 mg), which were also compared with extracts of *H. perforatum*, such as STW3-IV (612 mg), LI 160 (900 mg daily), and STW3-IV (900 mg), thereby obtaining a similar antidepressant efficacy but with fewer side effects, depending on the extract, as can be seen in Figure 1, thus being a better option when compared to the synthetic drugs [76,79,85,86].

*H. perforatum*, represented in any of its extracts, showed an antidepressant response equal to and even superior to synthetic drugs, which was based on the percentage of responders on the Hamilton scale, where the extracts ZE 117 (69%) and STWE (69%) stood out (66%), as well as the drugs imipramine (67%) and sertraline (60%). In contrast to this, the comparison with the presence of side effects was notoriously lower in the natural extracts than in the synthetic drugs, again highlighting the ZE 117 extract, which presented side effects in only 19% of its study population in contrast to 32% for sertraline, which was the drug that presented fewest adverse effects [73,74,75,76,86]. However, more studies are needed that simultaneously evaluate more than one drug in comparison with more than one *H. perforatum* extract under a defined selection protocol and in a broader population. It would also be preferred to carry out these studies in Chile, since there are no updated studies on the subject and they would be adjusted to the specific population of the country. Furthermore, additional studies advocating for St. John’s Wort as an alternative treatment for mild to moderate depressive disorders would be beneficial. The way that studies are conducted may also have an impact on the understanding of drug– *H. perforatum* interactions. This is particularly relevant in Chile, where drug usage rises significantly with age, alongside the common occurrence of polypharmacy (the use of five or more medications). The average number of drugs consumed is 1.4, with some individuals using as many as 16. Of these, 22.7% are analgesics, 13.3% are antihypertensive agents, and 8.3% are anti-inflammatory and antirheumatic medications [105].

Most of these drugs have presented an interaction with *H. perforatum*, which may affect the patient’s previous treatment, from altering the anticoagulant activity [106,107], greater bleeding in women between periods [108], increased lipids in sera [109], and altered cardiovascular therapies [110], to some of the most serious, including inducing failures in antiretroviral therapies [104,111], transplant rejections [102,112], and a decrease in the effects of chemotherapeutic drugs [113,114,115].

One of the most significant findings related to these interactions pertains to preparations with low levels of hyperforin, ranging from 0.1% to 0.3% of the H. perforatum extract. These preparations contain a reduced amount of hyperforin, which does not stimulate cytochromes or P-glycoprotein. As a result, the pharmacokinetics of drugs co-administered with the herb would remain unaffected [108,110,116]. Although there is a lack of studies to support these results, and to confirm what the maximum dose before an induction in metabolic pathways should be, these extracts with low levels of hyperforin could be the solution to avoiding possible interactions with other concomitant medications, and thus to providing a safer treatment for the patient.

A limitation of this study is the limited research available on the components that are present in smaller quantities within the herb, as well as the scarce information regarding the specific antidepressant activity of these components when considered individually. Additionally, it remains unclear whether these components act as enhancers or inhibitors of hypericin and hyperforin, whose effects are already well-established. Moreover, current studies have not adequately compared the traditionally prescribed medications for mild to moderate depressive disorder with the various extracts of H. perforatum while controlling for dosage proportions, treatment durations, and the scales used to assess the disorder, as well as the potential improvements observed at the conclusion of the studies. Additionally, many of the more complex studies suffer from issues related to their age, higher population densities, and a lack of detailed information about the composition of the extracts (Table 1).

As for the study of the side effects, there is no standard for these, nor is there minute detail provided about the method used by most of the studies aimed at them. Concerning the co-administration of other drugs, not all studies provide a detailed description of the variables that affect the pharmacokinetics of the medications, or the specific type of extract used, as well as their components. This information is crucial, especially in the time following an observed difference that may result from their interactions.

Due to the information collected, it is expected that this study will be considered when seeking an alternative treatment for mild-to-moderate depression disorder, in considering *H. perforatum* as a safe and efficient option, and when taking into consideration the possible interactions with drugs administered for other pathologies, in addition to promoting new studies to update and complement the information already available on St. John’s Wort and its antidepressant capacity.

## 5. Conclusions

Although the studies are conclusive, more research is needed on the components of *H. perforatum* to arrive at a detailed mechanism of its antidepressant effect, as well as the types of extracts that can be used for this purpose. *H. perforatum*, emerges as a promising natural alternative for the treatment of mild-to-moderate depression, demonstrating a notable antidepressant effect with significantly fewer side effects when compared to conventional synthetic antidepressants. This review highlights the key active compounds, hypericin and hyperforin, which contribute to its therapeutic efficacy while presenting a lower overall incidence of adverse effects—27% for *H. perforatum* compared to 43% for synthetic drugs, and only 19% for the specific extract labeled as ZE 117. However, it is crucial to approach its use with caution, as potential interactions with other medications may occur, underscoring the importance of medical supervision in its administration. This analysis highlights *H. perforatum* as a viable option for patients seeking alternative treatments, provided it is used under professional guidance. However, the need for caution cannot be overstated. Self-medication with this herb poses significant risks due to its active compound, which, while effective, is not without side effects. *H. perforatum* has the potential to interact with various medications—including oral contraceptives, anticoagulants, immunosuppressants, and certain antidepressants—potentially diminishing their effectiveness or causing serious adverse effects. To mitigate these risks, it is essential to consult a healthcare professional before initiating treatment. Medical supervision ensures that the herb is used safely and appropriately, helping patients understand its limitations and the necessary precautions to prevent complications. Furthermore, this underscores the importance of continued research to deepen our understanding of its benefits and risks, ensuring its safe and effective integration into therapeutic practices.

## Figures and Tables

**Figure 1 pharmaceuticals-17-01625-f001:**
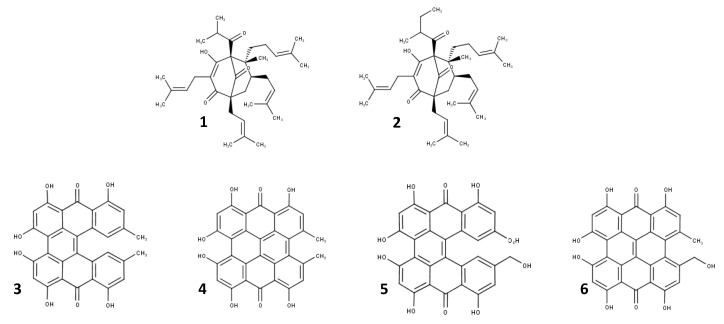
Structure of the molecules involved in the antidepressive activities found in *H. perforatum*. **1**: hyperforin; **2**: adhyperforin; **3**: protohypericin; **4**: hypericin; **5**: protopseudohypericin; **6**: pseudohypericin.

**Figure 2 pharmaceuticals-17-01625-f002:**
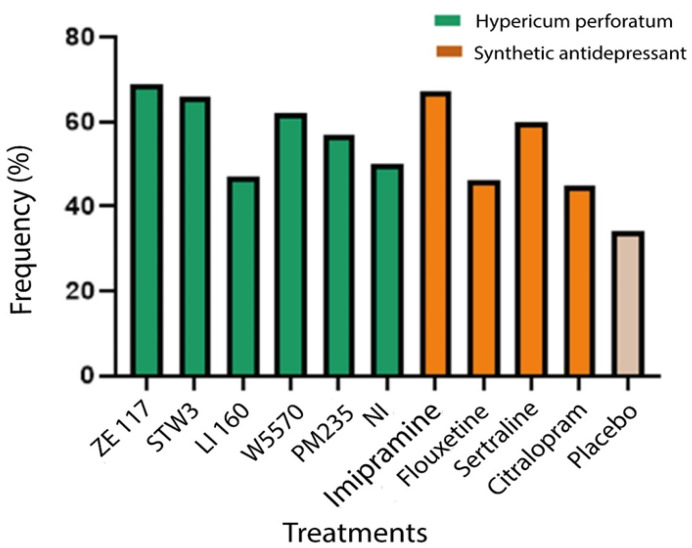
Treatment with synthetic antidepressants and extracts of *Hypericum perforatum*. Comparison of response rates among patients treated with St. John’s Wort and its various extracts versus synthetic drugs and the placebo based on the percentage of responders. Data are expressed as the proportion of patients who experienced a reduction of at least 50% of their initial Hamilton Depression Rating Scale (HAMD-17) scores after the specified treatment duration. This analysis highlights the relative efficacy and acceptance of each treatment option among the individuals who utilized them.

**Figure 3 pharmaceuticals-17-01625-f003:**
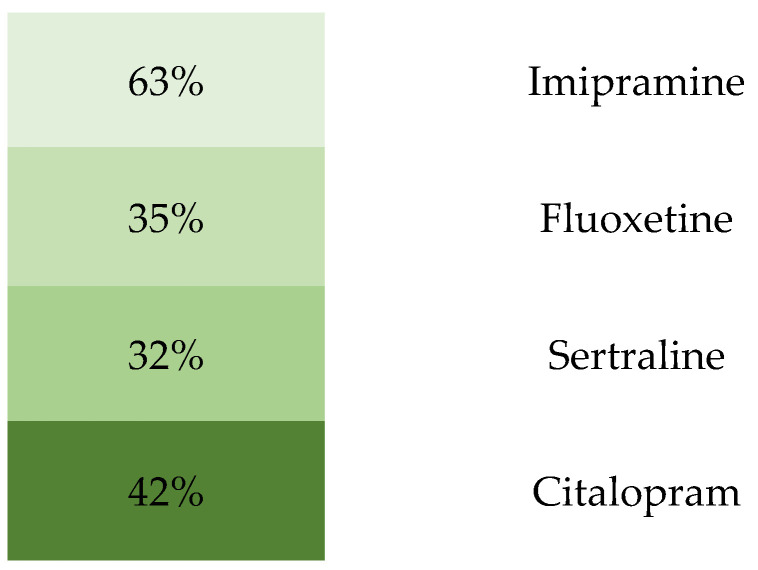
Adverse events in the treatments with synthetic antidepressants. The data reveal a high prevalence of side effects among patients treated with these medications. Imipramine exhibited the highest incidence of adverse effects at 63%, followed by citalopram at 42%. Sertraline, while having the lowest incidence among the three at 32%, still demonstrates a noteworthy percentage of patients experiencing side effects. This figure underscores the variability in the side-effect profiles across different synthetic antidepressants.

**Table 1 pharmaceuticals-17-01625-t001:** Components of *H. perforatum* and its antidepressant mechanisms.

Group	Components	Antidepressant Mechanism
Phloroglucinol derivatives	Hyperforin Adhyperforin	Inhibition of reuptake and integration with catecholamine receptors
Naphthodianthrones	Hypericin Pseudohypericin Isohypericin	Inhibition of MAOs

**Table 2 pharmaceuticals-17-01625-t002:** Composition of *H. perforatum* extracts.

Extracts	Hypericin (%)	Hyperforin (%)	References
ZE 117	Negligible	1–4	[18]
WS 5570	0.12–0.28	5–6	[19]
STEI 300	0.2–0.3	2–3	[20]
LI 160	0.3	Less than 0.1%	[21]

**Table 3 pharmaceuticals-17-01625-t003:** Efficacy of the different *H. perforatum* extracts for a total of 2004 patients, using the type of extract and dose administered to evaluate its efficacy.

Extracts	Patients	Dose (mg)	References
ZE 117	849	500	[72,73,74,75]
STW3	324	600–900	[76,77]
LI 160	112	600–900	[78,79,80]
W5570	473	300–1200	[13,19,81]
PM235	100	270	[82]
NI *	146	600–900	[83,84,85,86]
Total		2004	

NI * = non-specific *Hypericum perforatum* extract.

**Table 4 pharmaceuticals-17-01625-t004:** Patients treated with synthetic antidepressants.

Antidepressant	Patients	Dose	References
Imipramine	167	150	[75]
Fluoxetine	380	20	[73,78,83,84]
Sertraline	176	50–100	[76,79,86]
Citalopram	154	20	[87]
Total	877		

**Table 5 pharmaceuticals-17-01625-t005:** Comparison of side effects: *H. perforatum* vs. standard antidepressants. The table highlights that *H. perforatum* generally has fewer side effects compared to standard antidepressants, including SSRIs and tricyclic antidepressants. Multiple studies indicate a lower incidence of adverse events and dropout rates due to side effects for patients treated with *H. perforatum*. This makes it a potentially safer alternative for the treatment of depression, particularly for those concerned about the side effects of conventional antidepressants.

Study Title	Key Findings of Side Effects	References
St. John’s Wort for major depression	Patients given *H. perforatum* dropped out due to adverse effects less frequently than those given older antidepressants (OR 0.24; 95%CI; 0.13 to 0.46) or SSRIs (OR 0.53; 95%CI; 0.34 to 0.83)	[88]
St John’s wort for depression: an overview and meta-analysis of randomised clinical trials	Side effects occurred in 19.8% of patients on St John’s Wort compared to 52.8% on standard antidepressants	[89]
St John’s wort for depression: Meta-analysis of randomised controlled trials	The proportions of patients reporting side effects were 26.3% for *H. perforatum* vs. 44.7% for standard antidepressants	[90]
Safety of St. John’s Wort extract compared to synthetic antidepressants	Adverse effect incidence for *H. perforatum* ranged from 0 to 6%, which was significantly lower than for synthetic antidepressants	[91]
St John’s wort superior to placebo and similar to antidepressants for major depression but with fewer side effects	*H. perforatum* extracts are effective for major depression and have significantly fewer risks compared to synthetic antidepressants	[92]
St John’s wort: Prozac from the plant kingdom	*H. perforatum* is associated with fewer adverse reactions compared to conventional antidepressants	[93]
St. John’s wort: a new alternative for depression?	*H. perforatum* has fewer cardiac effects and anticholinergic side effects compared to tricyclic antidepressants and monoamine oxidase inhibitors	[94]
St. John’s wort and depression: Efficacy, safety and tolerability-an update	*H. perforatum* is comparable to other antidepressants in efficacy but lacks major side effects, making it a safer option	[95]
St. John’s Wort: A Systematic Review of Adverse Effects and Drug Interactions for the Consultation Psychiatrist	Generally well tolerated with fewer side effects compared with tricyclic antidepressants in the short-term management of mild-to-moderate depression	[96]

**Table 6 pharmaceuticals-17-01625-t006:** Comparative table of side effects: *H. perforatum* vs. synthetic/traditional antidepressants.

Symptom/Side Effect	St. John’s Wort (%)	Synthetic/Traditional Antidepressants (%)	Comments
Nausea	5–10%	15–30%	Less frequent with St. John’s Wort
Fatigue	10–15%	20–35%	Generally better tolerated
Insomnia	5–10%	10–20%	Similar frequency
Dry mouth	5–10%	15–25%	More common with antidepressants
Dizziness	5–12%	15–30%	Less reported with St. John’s Wort
Weight gain	Rare (<5%)	20–30%	More prevalent with antidepressants
Sexual dysfunction	Rare (<5%)	30–50%	Very common with SSRIs and standard antidepressants
Anxiety or agitation	10%	15–25%	Slightly lower with St. John’s Wort
Gastrointestinal issues	5–10%	20–30%	More common with antidepressants
Headache	5–10%	15–25%	Less frequent with St. John’s Wort
Excessive sedation	Rare (<5%)	10–20%	More common with sedative antidepressants
Drowsiness	5–10%	20–30%	More prevalent with antidepressants
Excessive sweating	5–10%	15–25%	More reported with antidepressants
Confusion or disorientation	Rare (<5%)	10–15%	Less common with St. John’s Wort
Tremors	Rare (<5%)	5–10%	Similar frequency
Phototoxicity	5–10%	Rare (<5%)	Exclusive to St. John’s Wort in some cases
Tachycardia	Rare (<5%)	5–10%	Similar frequency
Hypotension	Very rare (<1%)	5–10%	More common with antidepressants

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
