# Peer review of "Documentary Analysis of Hypericum perforatum (St. John’s Wort) and Its Effect on Depressive Disorders"

_pharmaceuticals, 2024, doi:10.3390/ph17121625_

Round 1

Reviewer 1 Report

Comments and Suggestions for Authors

Dear Authors, thank you for submitting an interesting article entitled Documentary analysis of Hypericum perforatum (St. John's Wort) and its effect on depressive disorders” because the transfer of information is always useful.

Dear Authors, thank you for submitting an interesting article entitled ”Documentary analysis of Hypericum perforatum (St. John's Wort) and its effect on depressive disorders” because the transfer of information is always useful.
Below you can find some general and specific comments.

The article aims to assess the antidepressant efficacy of Hypericum perforatum (St. John's Wort) as a potential natural alternative to synthetic antidepressants. By synthesizing findings from clinical trials and literature from early 2020, the authors highlight that H. perforatum may provide benefits in treating mild to moderate depression with fewer side effects than synthetic antidepressants.

The article emphasizes the active compounds hypericin and hyperforin and reviews potential mechanisms of action, as well as the safety profile and lower adverse event incidence of H. perforatum, particularly the ZE 117 extract. The review provides a valuable overview for medical professionals considering complementary therapies for depression.

General Comments

The manuscript is well-structured, presenting a clear rationale for exploring H. perforatum as an alternative treatment for depression. The authors appropriately discuss its potential to reduce side effects relative to synthetic antidepressants, which adds relevance to the field of psychopharmacology.
The article effectively discusses key active compounds and their proposed mechanisms, which contributes to a deeper understanding of how H. perforatum may exert antidepressant effects.

The discussion of adverse effects is a valuable addition, especially the comparison between H. perforatum and synthetic antidepressants. The authors effectively interpret the findings regarding side effects, providing a comparison of specific adverse events for H. perforatum versus synthetic treatments.

Specific Comments
Line 38: please organize the keywords in alphabetical order and italicize Hypericum perforatum
Line 48: please write the DSM-V abbreviation in full detail
Lines 386, 396, 452, 547: please italicize Hypericum perforatum
Lines 393-394: From figure 5 I would like to see not only the side effects but if there were statistically significant differences. How was the graph made? Based on what data? At a first analysis it can be seen that they have the same side effects, and in some cases even much greater.
Line 403: Erp: Eruption; Dol: Pains; Trans: Disorders; Disf: Dysfunction; Enf: Disease - where do they come from?

Please the authors to underline in the text, in the discussion and or conclusions why is the use of Hypericum perforatum an alternative? What is the basis for this claim (price, side effects, accessibility, compounds etc)? Of course, data must be supported by a statistical analysis, not just numerical or percentage.
The authors mention a bibliographic review of clinical trials but do not detail the number of studies included. Including specific numbers and study types would help clarify the breadth and depth of this review (We found 945 results, which 139 have been revised with the following inclusion criteria in
mind……. References 114).

The manuscript states, "St. John's wort extracts proved to be a safer option in patients, presenting only 27% of adverse effects compared to synthetic antidepressants," but the data source for these percentages is unclear.
The review provides numerical comparisons of adverse effect rates, which is essential for the reader's understanding.

The conclusions align with the evidence provided regarding the reduction of adverse effects and potential for mild to moderate depression treatment.
There is a need for cautionary language about self-medication risks, given the interactions of H. perforatum with other medications. Emphasizing the importance of medical supervision might strengthen this point, ensuring readers understand the limitations and necessary precautions of using H. perforatum.

Best regards!

Author Response

Thank you for your thoughtful feedback regarding our manuscript. We appreciate your comments and the opportunity to clarify some points related to our findings.

Line 38: please organize the keywords in alphabetical order and italicize Hypericum perforatum

R: We added Keywords in line 36. Hypericum perforatum is now italicized throughout the text

Line 48: please write the DSM-V abbreviation in full detail

R: done as suggested

Lines 386, 396, 452, 547: please italicize Hypericum perforatum

R: done as suggested

Lines 393-394: From figure 5 I would like to see not only the side effects but if there were statistically significant differences. How was the graph made? Based on what data? At a first analysis it can be seen that they have the same side effects, and in some cases even much greater.

R: Figure was very confused, so we decided to make a table with stronger data with its respective studies. We would like to clarify that there is currently no standardized protocol for assessing adverse symptoms, which contributes to variability in the literature. Additionally, not all articles fully report or track adverse symptoms or sometimes do not report their incidence frequency, resulting in gaps in the data. These issues hinder our ability to conduct statistical analyses, as the inconsistent reporting makes meaningful comparisons difficult.

Line 403: Erp: Eruption; Dol: Pains; Trans: Disorders; Disf: Dysfunction; Enf: Disease - where do they come from?

R: Data was deleted from the graph

Please the authors to underline in the text, in the discussion and or conclusions why is the use of Hypericum perforatum an alternative? What is the basis for this claim (price, side effects, accessibility, compounds etc)? Of course, data must be supported by a statistical analysis, not just numerical or percentage.

R: In conclusion, the use of Hypericum perforatum may be an alternative to the traditional use of antidepressant drugs because; first, it has an effect on depression with specific active compounds such as hypericin and hyperforin, for which molecular mechanisms have been described. Second, it shows a significantly lower incidence of side effects compared to synthetic antidepressants. Furthermore, Hypericum perforatum has also been described as having traditional use in regions like Europe as an antidepressant, an effect noted in European ethnopharmacology, which implies easy and cheap accessibility for the population. These three factors strongly indicate that Hypericum perforatum may serve as a viable alternative to synthetic antidepressants, particularly for individuals looking for options with minimal side effects.

The authors mention a bibliographic review of clinical trials but do not detail the number of studies included. Including specific numbers and study types would help clarify the breadth and depth of this review (We found 945 results, which 139 have been revised with the following inclusion criteria in mind……. References 114).

R: From all discussed articles, only 16 articles include randomized controlled trials comparing the effectiveness of St. John's Wort with other medications, including standard antidepressants, in the treatment of depression. This information was included in the manuscript.

The manuscript states, "St. John's wort extracts proved to be a safer option in patients, presenting only 27% of adverse effects compared to synthetic antidepressants," but the data source for these percentages is unclear.

The review provides numerical comparisons of adverse effect rates, which is essential for the reader's understanding.

R: two sentences were replaced, one in the abstract and one in results, instead a more detail explanation was given.

The conclusions align with the evidence provided regarding the reduction of adverse effects and potential for mild to moderate depression treatment.

There is a need for cautionary language about self-medication risks, given the interactions of H. perforatum with other medications. Emphasizing the importance of medical supervision might strengthen this point, ensuring readers understand the limitations and necessary precautions of using H. perforatum.

R: done as suggested

Reviewer 2 Report

Comments and Suggestions for Authors

The authors evaluated the potential of Hypericum perforatum as a natural alternative to synthetic antidepressants for treating depression, highlighting its comparable efficacy, highlighting its comparable efficacy with fewer adverse effects—27% for Hypericum perforatum extracts versus 43% for synthetic antidepressants. In general, the main areas for improvement are the presentation of tables and figures. Specific comments are lay out below: 

·       Please double check some grammar issues. For example, on line 29-33 in the Abstract section, there seems to be a cut off after “among which %”.

·       Please remove the extra row at the bottom of Table 2.

·       In Table 3, please add the explanation of NI* right under the table, though it was found later on line 347. 

·       Please consider using a table to demonstrate the data in Figure 5. The current presentation is hard to follow as there are too many information on the x axis.

Author Response

Please double check some grammar issues. For example, on line 29-33 in the Abstract section, there seems to be a cut off after “among which %”.

R: Paragraph has been improved and replaced.

Please remove the extra row at the bottom of Table 2.

R: done as suggested

In Table 3, please add the explanation of NI* right under the table, though it was found later on line 347.

R: done as suggested

Please consider using a table to demonstrate the data in Figure 5. The current presentation is hard to follow as there are too many information on the x axis.

R: Yes, we totally agree. We decided to include two tables instead. One table with the comparison of side effects: St. John's Wort vs. standard antidepressants and the other table exposing which side effects.